# Isolation and Characterization of Nanocellulose with a Novel Shape from Walnut (*Juglans Regia* L.) Shell Agricultural Waste

**DOI:** 10.3390/polym11071130

**Published:** 2019-07-03

**Authors:** Dingyuan Zheng, Yangyang Zhang, Yunfeng Guo, Jinquan Yue

**Affiliations:** 1College of Material Science and Engineering, Northeast Forestry University, Harbin 150040, China; 2Key Laboratory of Bio-based Material Science and Technology of Ministry of Education, Northeast Forestry University, 26 Hexing Road, Harbin 150040, China

**Keywords:** Walnut shell, nanocellulose, TEMPO oxidation, sulfuric acid hydrolysis, ultrasonication

## Abstract

Herein, walnut shell (WS) was utilized as the raw material for the production of purified cellulose. The production technique involves multiple treatments, including alkaline treatment and bleaching. Furthermore, two nanocellulose materials were derived from WS by 2,2,6,6-tetramethylpiperidine-1-oxyl radical (TEMPO) oxidation and sulfuric acid hydrolysis, demonstrating the broad applicability and value of walnuts. The micromorphologies, crystalline structures, chemical functional groups, and thermal stabilities of the nanocellulose obtained via TEMPO oxidation and sulfuric acid hydrolysis (TNC and SNC, respectively) were comprehensively characterized. The TNC exhibited an irregular block structure, whereas the SNC was rectangular in shape, with a length of 55–82 nm and a width of 49–81 nm. These observations are expected to provide insight into the potential of utilizing WSs as the raw material for preparing nanocellulose, which could address the problems of the low-valued utilization of walnuts and pollution because of unused WSs.

## 1. Introduction

With the depletion of fossil fuels and the increase in ecological and environmental problems caused by the usage of fossil fuels, the development and utilization of green biomass-based materials derived from renewable natural resources have been extensively studied around the world [1]. Cellulose, which can be observed in all the plant structures, is a common example of a renewable natural resource and offers the advantages of abundant availability, renewability, and biodegradability [2,3]. With the development of nanotechnology, nanocellulose, prepared from cellulose, has attracted significant attention from academic and industrial researchers because of its low cost, biocompatibility, biodegradability, nontoxicity, renewability, sustainability, strong surface reactivity, and desirable physical properties (it is lightweight and impermeable to gas and it also exhibits high stiffness, good optical transparency, and low thermal expansion) [4,5]. Compared with cellulose, nanocellulose exhibits large surface area, high crystallinity, high mechanical strength, high hydrophilicity and supramolecular structure [6]. These characteristics make nanocellulose promising for various applications such as polymer nanocomposites [7,8], packaging [9,10], electronics [11,12], and stimulus-responsive materials [13,14].

Nanocellulose can be extracted from several cellulose resources such as wood [15,16], cotton [17,18], ramie [19], bagasse [20,21], bamboo [22], sisal [23], corn straw [24], rice straw [18], and coconut shell [25]. It has been reported that the fundamental properties of the obtained nanocellulose, such as morphology, crystallinity, dimensions, and surface chemistry, vary highly depending on the raw material and the isolation process used to obtain it [26,27]. Hence, different categories of nanocellulose, such as cellulose nanofibrils, cellulose nanocrystals, and bacterial cellulose, which differ in terms of their dimensions and morphologies, exist [2,28]. These key properties are of critical importance for ensuring the end use of the isolated nanocellulose.

Walnut (*Juglans regia* L.), also known as a peach, is harvested from the seed of the walnut tree. Walnut is one of the world’s four major “dried fruits” along with almonds, cashews, and hazelnuts. The global walnut production in 2017 exceeded 3.8 million tons. The walnut shell (WS) is the hard outer shell (endocarp) of walnut, which comprises cellulose, hemicellulose, lignin, and other small molecular substances and accounts for 67% of the total weight of walnut [29]. China, which is the top walnut producer in the world, produced 1.92 million tons of walnuts in 2017, which resulted in approximately 1.29 million tons of WSs (UN Food and Agriculture Organization, Corporate Statistical Databas, 2018). After the nut is removed from the WS, the shells are considered to be agricultural and forestry waste, which is often burned as fuel, seriously damaging the environment. Only a small portion of the produced WSs is used for preparing wood–plastic composite materials [30], carbon materials [31,32], or handcrafted products. Thus, the utilization of WSs is not much valued. The walnut contains desirable substances such as cellulose, hemicellulose, and lignin. It has been reported that the cellulose content of walnut shell is about 22% [33]. Hence, in this study, we investigate the preparation of nanocellulose using WS as the raw material for the first time.

Cellulose nanofibrils are mainly obtained from cellulosic fibers by mechanical treatments such as high-pressure homogenization, microfluidization, grinding, and ultrasonication [34]. However, such mechanical fibrillation methods are very energy intensive. Various pretreatment methods (such as enzymatic hydrolysis [35], 2,2,6,6-tetramethylpiperidine-1-oxyl radical (TEMPO)-mediated oxidation [36,37,38]) have been proposed to reduce the energy required for the mechanical deconstruction process by reducing the negative or positive charge on the fiber surfaces and by enhancing the colloidal stability of the final cellulose nanofibrils.

Unlike cellulose nanofibrils, cellulose nanocrystals have a rod-like morphology [39]. Cellulose nanocrystals are generally prepared by hydrolysis using a strong inorganic acid such as sulfuric acid [40,41], other inorganic strong acid hydrolysis are used for cellulose nanocrystals preparation as well, including hydrochloric acid, phosphoric acid, hydrobromic acid, and nitric acid [42]. Recent technological developments have resulted in a few sustainable and environment friendly methods that rely on recyclable chemicals. Examples include hydrolysis with solid acids (e.g., phosphor tungstic acid [43]) or treatment with ionic liquids [44] or deep eutectic solvents [45]. Among these, sulfuric acid is the most commonly used acid for producing sulfonated cellulose nanocrystals exhibiting good dispersibility in water [42]. During the hydrolysis process, the paracrystalline or disordered parts of cellulose are hydrolyzed and dissolved in the acid solution; however, the crystalline parts are chemically resistant to the acid and remain intact. Consequently, the cellulose fibrils are transversely cleaved, yielding short cellulose nanocrystals with a relatively high crystallinity [34].

In this study, we aim to isolate nanocellulose from a renewable, cheap, and currently underutilized raw material, i.e., WS. Herein, WS is pretreated using several mechanical and chemical processes, including grinding, extraction, alkali treatment, and bleaching. Nanocellulose was produced by TEMPO oxidation accompanied by ultrasonic treatment (yielding TNC) or sulfuric acid hydrolysis followed by ultrasonication (SNC). The resulting TNC and SNC and the intermediate products were characterized by transmission electron microscopy (TEM), X-ray diffraction (XRD), Fourier-transform infrared spectroscopy (FTIR), and thermogravimetric analysis (TGA). The proposed method of extracting nanocellulose from WSs is expected to reduce the pressure on natural resources. This will offer an alternative high-valued utilization for the currently wasted WSs and, furthermore, to serve as a reference for future studies to improve the utilization of the obtained nanocellulose for developing new biobased nanomaterials.

## 2. Materials and Methods

### 2.1. Materials

The WS collected from the Shanxi Province in China was used as the raw material. Ethanol, sodium hydroxide, sodium chlorite, sodium bromide, sodium hypochlorite, glacial acetic acid, sulfuric acid, and other chemicals were of analytical grade and used without any further purification. All of the chemicals were supplied by Tianjin Kemiou Chemical Reagent Co., Ltd. (Tianjin, China). Distilled water was used the whole process. TEMPO was purchased from Aladdin Reagent Co., Ltd. (Shanghai, China). Dialysis bags (MD44, viskase, Lombard, IL, USA) were provided by Beijing Biotopped Science & Technology CO., Ltd. (Beijing, China).

### 2.2. Preparation of Nanocellulose

The procedure used to prepare nanocellulose is outlined in Figure 1.

#### 2.2.1. WS Pretreatment

In natural WSs, cellulose is embedded in a network structure in which lignin and hemicellulose are linked together, mainly by covalent bonds [46]. Thus, WS was initially pretreated to isolate the nanocellulose. Pretreatment of walnut shell was performed using the method described by de Rodriguez, Thielemans, and Dufresne [47]. The WS was ground and sieved through 60 mesh (0.25 mm), dried in an oven at 105 °C, and stored in a desiccator. To eliminate hemicellulose, the obtained WS powder was mixed with a 2 wt.% NaOH solution in a WS/NaOH solid-to-liquid ratio of 10 g/100 mL and stirred for 4 h at 100 °C. This NaOH treatment was repeated four times until no more discoloration occurred; the product obtained after the NaOH treatment can be referred to as N-WS. Further, N-WS was bleached in a solution containing equal amounts of acetate buffer and 1.7 wt.% NaClO_2_ at a WS/solution solid-to-liquid ratio of 5 g/100 mL for 6 h at 80 °C (the reaction flask was shaken every 20 min to ensure the reaction occurs evenly). This bleaching process was repeated four times. The bleached products were then thoroughly washed using distilled water. The obtained product can be referred to as B-WS.

#### 2.2.2. TNC Preparation

TNC was prepared using the method described by Kuramae, Saito, and Isogai [48], with few modifications. One gram of B-WS was dispersed in 100 mL of distilled water; further, 0.1 g of sodium bromide and 0.016 g of TEMPO were added, followed by 20 mmol of NaClO. During the TEMPO oxidation, the pH of the suspension was adjusted to 10 ± 0.5 using 0.1 M NaOH and 0.1 M HCl. The reaction was continued for 5 h at room temperature under magnetic stirring at 1000 rpm and was terminated by adding 10 mL of ethanol. The oxidized B-WS was washed and filtered; subsequently, it was stored with distilled water at 4 °C to avoid strong hydrogen bonding.

To convert the nonoxidized hydroxyl and aldehyde groups in the oxidized B-WS into carboxyl groups, the product obtained in the previous step was further processed. One gram of oxidized B-WS (dry weight) was dispersed in 65 mL of distilled water, and the pH value was adjusted to 4–5. Further, 0.6 g of NaClO_2_ was added to the reaction system. The reaction was allowed to proceed for 1 h at 70 °C under magnetic stirring at 1000 rpm to yield carboxylated B-WS. The obtained carboxylated B-WS had a carboxylate content of 1.12 mmol/g as determined by conductivity titration [49]. A certain amount of carboxylated B-WS was weighed, and the mass fraction was adjusted to 0.5 wt.%. Finally, high-intensity ultrasonication was performed in an ultrasonic cell pulverizer (SCIENTZ-1200E, Ningbo Scientz Biotechnology Co., Ltd., Ningbo, China) at 600 w for 30 min in an ice/water bath to yield TNC.

#### 2.2.3. SNC Preparation

SNC was prepared using the method described by Beck-Candanedo [41]. One gram of B-WS was added to 8.75 mL of the sulfuric acid solution (64 wt.%) under vigorous stirring, and the hydrolysis reaction was allowed to proceed for 1 h at 45 °C. The reaction was ended by adding distilled water in a volume that was ten times the reaction volume. The obtained suspension was washed by mixing with distilled water followed by centrifugation at 12,000 rpm for 15 min to eliminate the excess acid; the washing was repeated until the precipitate generation was terminated. The suspension was subsequently dialyzed against distilled water until the pH became 6.5–7. The dialyzed suspension was sonicated at 600 w for 2 min to yield SNC.

### 2.3. Characterization of TNC, SNC, and Intermediate Products

#### 2.3.1. Analysis of the Chemical Components

The α-cellulose content was determined as follows [50]. Two grams of WS powder was weighed and transferred to a 250-mL Erlenmeyer flask, and 25 mL of nitric acid/ethanol solution (1:4 by volume) was added. The mixed solution was refluxed for 1 h, and the process was repeated several times until the sample turned white. The powder was repeatedly washed using distilled water and filtered until the pH became neutral; the obtained residue was dried at 105 °C. The α-cellulose content (*X*, %) was calculated as follows (1)$$X=\frac{G}{G_{1}\left(1-W\right)}\times 100$$ where *G* denotes the weight of the obtained residue, *G_1_* denotes the weight of the WS, and *W* denotes the moisture content of the WS.

The remaining chemical components of WS (organic extracts, lignin, hemicellulose, ash, and holocellulose) were analyzed according to the Technical Association of Pulp and Paper Industry standards that have been previously described [51].

#### 2.3.2. Scanning Electron Microscopy

The WS, N-WS, B-WS, TNC, and SNC microstructures were observed using a scanning electron microscope (SU8010, Hitachi, Japan) at an accelerating voltage of 5.0 kV. TNC and SNC were freeze-dried before observation. All the samples were coated with gold.

#### 2.3.3. Transmission Electron Microscopy

TNC and SNC were imaged using a transmission electron microscope (H-7650 Hitachi, Japan) at a 100-kV acceleration voltage. The TNC and SNC suspensions were diluted to a concentration of 0.01% and deposited onto carbon-coated grids (230 mesh, Beijing Zhongjingkeyi Technology Co., Ltd., Beijing, China). After drying, the samples were negatively stained using a 1% phosphotungstic acid solution for 10 min. The TEM images were analyzed using Nanomeasurer 1.2 (Department of Chemistry, Fudan Univ., Shanghai, China) to determine the TNC and SNC size distributions.

#### 2.3.4. Fourier-transform Infrared Spectroscopy

The FTIR spectra of WS, N-WS, B-WS, TNC, and SNC were recorded on a Fourier-transform infrared instrument (Nicolette 6700, Thermo Fisher Scientific Inc., Waltham, MA, USA) in 400–4000 cm^−1^ with a resolution of 4 cm^−1^, and 20 scans for each sample were conducted. The FTIR spectra of all samples were collected using the attenuated total reflection technique (ATR, the ATR crystal material is zinc selenide (ZnSe)).

#### 2.3.5. X-ray Diffraction Technique

XRD analysis was performed on WS, N-WS, B-WS, TNC, and SNC using a diffractometer (D/max 2200, Rigaku, Japan) equipped with Ni-filtered Cu Kα radiation (λ = 1.5406 Å) at 40 kV and 30 mA. The diffraction intensities were recorded in 2θ = 5°–60° with a scan rate of 5°/min.

The crystallinity index (CrI, %) was calculated according to the method reported by Segal [52] as follows (2)$$CrI=\frac{I_{200}-I_{am}}{I_{200}}\times 100$$ where *I_200_* is the maximum intensity of the diffraction at 200 peak (2θ = 22.6°) and *I_am_* is the intensity of the diffraction at 2θ = 18°.

#### 2.3.6. Thermogravimetric Analysis

The thermal stability of each sample was evaluated using a thermogravimetric analyzer (TG209F1, Netzsch Scientific Instruments Trading (Shanghai) Co., Ltd., Shanghai, China) from room temperature to 600 °C at a rate of 10 °C /min in a nitrogen atmosphere.

## 3. Results and Discussion

### 3.1. Chemical Components

Table 1 presents the WS, N-WS, and B-WS chemical compositions. The chemical compositions of the three samples were observed to be significantly different because of the applied chemical treatments.

It is found that WS has the lowest percentage of cellulose and highest percentage of noncellulosic components such as lignin and hemicelluloses. The chemical treatments aim to remove the noncellulosic components. When WS was subjected to NaOH treatment, the lignin and hemicelluloses contents of N-WS decreased to 30.98% and 7.6%, respectively, whereas the cellulose concentration of N-WS increased to 56.6%. After bleaching treatment, the cellulose content of B-WS increased to 87.9% due to the removal of remaining lignin and hemicelluloses, resulting in highly purified cellulose.

### 3.2. SEM Analysis

The WS, N-WS, B-WS, TNC, and SNC microstructures were observed using a scanning electron microscope. The SEM image of WS (Figure 2a) denotes that the WS had a rough surface. However, the surface of N-WS (as seen in Figure 2b) had an irregular porous structure due to the degradation of hemicellulose and the partial degradation of lignin by the repeated alkali treatment. Further, B-WS (shown in Figure 2c) exhibited a loose structure, which indicated the successful removal of residual lignin by the acetate buffer/NaClO_2_ solution. These results are consistent with the changes in chemical composition denoted in Table 1.

After the TNC was freeze-dried, the obtained aerogel was observed to be a porous network with a lamellar structure (Figure 2d). This structure is because the strong hydrogen bonding during freeze-drying caused the nanoparticles to self-assemble into the lamellar structures [53,54]. The aerogel formed by freeze-drying the SNC exhibited a similar porous structure (Figure 2e).

### 3.3. TEM Analysis

TEM was used to observe the morphology of the nanocellulose produced using different methods. The prepared TNC (Figure 3a) exhibited an irregular block structure, whereas the prepared SNC (Figure 3b) was rectangular with a length of 55–82 nm and a width of 49–81 nm. The TNC morphology observed in this study was significantly different from that of the nanocellulose produced using the same procedure in a previous study [36,54]; this may be attributed to the irregular morphological structure [30] of the WS powder used in this study, as depicted in Figure 2a. Furthermore, the SNC morphology obtained through sulfuric acid hydrolysis as observed by TEM differed from the typical rod- or needle-like morphologies of the typical cellulose nanocrystals [2]. As has been reported, the size and shape of nanocelluloses influence the properties (for example, optical characteristics, stability, and rheology) in aqueous media [55], which largely determines the application of nanocellulose. Nanocellulose with spherical or square structure makes them excellent candidates as stabilizer for Pickering emulsion [53] or drug delivery carrier for encapsulation [56].

### 3.4. Chemical Structures

The changes in the chemical structures of the raw WS after various treatments were investigated using FTIR (Figure 4). Two main absorption regions appeared in all the curves among which one was in the high-wave-number region from 2800 to 3500 cm^−1^ and the other was in the low-wave-number region from 600 to 1750 cm^−1^ [57]. The peak at 2897 cm^−1^ was attributed to the stretching vibration of the C–H groups of cellulose, whereas the wide region around 3350 cm^−1^ was attributed to the O–H stretching vibration of the hydrogen-bonded hydroxyl groups in the cellulose molecules [51,58]. Further, the peak at 1738 cm^−1^ corresponds to the acetyl groups and ironic esters of the hemicellulose and the ester linkages of the carboxylic groups of the ferulic and p-coumaric acid in lignin and hemicellulose [59]. The absence of a peak at 1738 cm^−1^ in the FTIR spectrum of N-WS indicates that the hemicellulose in WS was effectively degraded by the NaOH treatment. The peaks at 1248 cm^−1^ and 1502 cm^−1^ in the FTIR spectrum of WS, corresponding to the aromatic skeletal vibrations of lignin [60], disappeared in the curve of B-WS, confirming that majority of the lignin was removed by the bleaching treatment. Furthermore, the absence of peaks at 1738 cm^−1^, 1502 cm^−1^, and 1248 cm^−1^ in the B-WS spectrum is consistent with the changes in chemical components during the alkaline and bleaching processes. The absorption peaks at 1162 cm^−1^ and 1034 cm^−1^ in all the curves correspond to the stretching vibration of the C–O–C bonds in the 1,4-glycosidic links linkages of the d-glucose units in cellulose, which were interpreted as typical for a cellulose structure [61]. The peaks at approximately 1645 cm^−1^ in the spectra of all the samples were attributed to the H–O–H stretching vibration of the adsorbed water due to the hydroxyl groups in cellulose [20], whereas those at 892 cm^−1^ represented the C_1_–H deformation of cellulose [62]. There were no significant differences between the SNC and B-WS spectra, demonstrating that the characteristics of the cellulose molecular structure were maintained during sulfuric acid hydrolysis. The increase in the relative amount of cellulose in the sample due to the decrease in the amounts of other components upon hydrolysis may account for the slight increase in the intensity of the peak at 1034 cm^−1^ from B-WS to SNC. The peak at 1738 cm^−1^, which is characteristic of the C=O stretching of carboxyl groups, reappeared in the FTIR spectrum of TNC when compared with that of B-WS due to the introduction of –COOH on the cellulose surface; this indicates that cellulose was successfully modified by the TEMPO oxidation [63].

### 3.5. Crystal Structures

The XRD patterns of WS, N-WS, B-WS, TNC, and SNC (Figure 5) were studied to further evaluate the influence of the processing treatment. All the samples exhibited diffraction peaks at approximately 16° (110) and 22.6° (200), whereas WS, N-WS, and B-WS also exhibited a diffraction peak at approximately 34° (004). These peaks indicate that the typical cellulose crystal structure was preserved in all the samples, indicating that the chemical and ultrasonic treatments did not change the integrity of the original cellulose crystal [64,65]. However, the crystallinity index changed with each step. The apparent crystallinity of WS was 29.5%. After NaOH treatment, the CrI of N-WS increased to 40.9% due to the dissolution and removal of lignin and majority of the hemicellulose (i.e., the amorphous hemicellulose) [57]. The CrI further increased to 42.9% as a result of the lignin removal by the acetate buffer/NaClO_2_ solution. These results are consistent with the observed changes in the chemical composition and the FTIR analyses. The CrI of SNC (40.1%) decreased slightly after the sulfuric acid hydrolysis and ultrasonication, which may have been caused because the amorphous and crystalline regions were damaged by the strong acid hydrolysis [66]. There were no significant differences in the position of X-ray diffraction peaks between the TNC and B-WS XRD curves, indicating that the original crystal structure of the cellulose was unchanged after the oxidation (Figure 5). These results indicate that the carboxylate groups formed by the TEMPO-mediated oxidation are selectively introduced on the surfaces of the cellulose microfibril rather than the internal cellulose crystallites [38].

### 3.6. Thermal Stability

Figure 6 and Figure 7 depicted the TGA and DTG curves, respectively, for WS, N-WS, B-WS, TNC, and SNC. All the TGA curves began with slight mass losses from room temperature to 105 °C, corresponding to the evaporation of slightly bound water from all the samples. As the temperature increased further, the WS degradation occurred in two phases. The first decomposition occurred between 214 °C and 300 °C, corresponding to the hemicellulose degradation and the beginning of lignin degradation [23,67]. After this change, the largest loss of mass in the material occurred between 300 °C and 380 °C, peaking at 342 °C, which corresponded to that of cellulose [68].

In contrast, the decomposition of four other samples occurred in only one stage. N-WS began to degrade at 205 °C due to lignin degradation; however, the maximum mass loss occurred at 301 °C, which can be attributed to cellulose degradation. The absence of peaks related to the degradation of hemicellulose and the relatively small change associated with lignin degradation in this sample as compared with that associated with WS indicates that the alkaline treatment eliminated most of the hemicellulose and some of the lignin from the WS.

B-WS degraded from 222 to 380 °C, with the most significant mass loss occurring at 332 °C; this was mainly attributed to cellulose degradation, which was within the decomposition temperature range reported in previous studies [68,69] (315–400°C). The wide range, which was associated with lignin degradation, was not observed in this sample, indicating that lignin was eliminated after the bleaching reaction in the acetate buffer/NaClO_2_ solution.

The thermal degradation of TNC (i.e., following TEMPO/NaClO/NaBr oxidation) began at 178 °C, which was approximately 50 °C lower than that in B-WS. It has been reported that the introduction of –COOH by the TEMPO-mediated oxidation of the C6 primary hydroxyl groups on the cellulose surface results in a significant decrease in thermal degradation [70].

SNC exhibited the worst thermal stability among the five tested samples, which can be attributed to the introduction of sulfate groups. It had been reported that the degradation of sulfate groups requires relatively low activation energy [71,72]. The thermal stability of TNC was likely to be lower than that of SNC because it exhibited fewer crystalline regions. This result is consistent with those in previous studies [73,74], which have shown that the crystalline regions of nanocellulose generally provide thermal stability. This finding is in agreement with the finding of XRD analyses, which showed that TNC had a high crystallinity index.

## 4. Conclusions

In this study, nanocellulose was produced using WS as the raw material; novel shapes were obtained by applying different production processes. The chemical composition analysis revealed that the hemicellulose and lignin in WS were effectively removed by alkali treatment and bleaching, increasing the cellulose content to 89%. The TEM images denoted that the produced TNC exhibited an irregular block structure, whereas SNC was rectangular with a length of 55–82 nm and a width of 49–81 nm. The results of the FTIR and XRD analyses of WS, N-WS, and B-WS were consistent with the observed chemical changes and confirmed that the typical cellulose structure remained unchanged through the TEMPO oxidation and sulfuric acid hydrolysis treatments. Our observations demonstrate that WSs, which are an abundant and sustainable agricultural waste, may be repurposed for the production of nanocellulose.

## Figures and Tables

**Figure 1 polymers-11-01130-f001:**
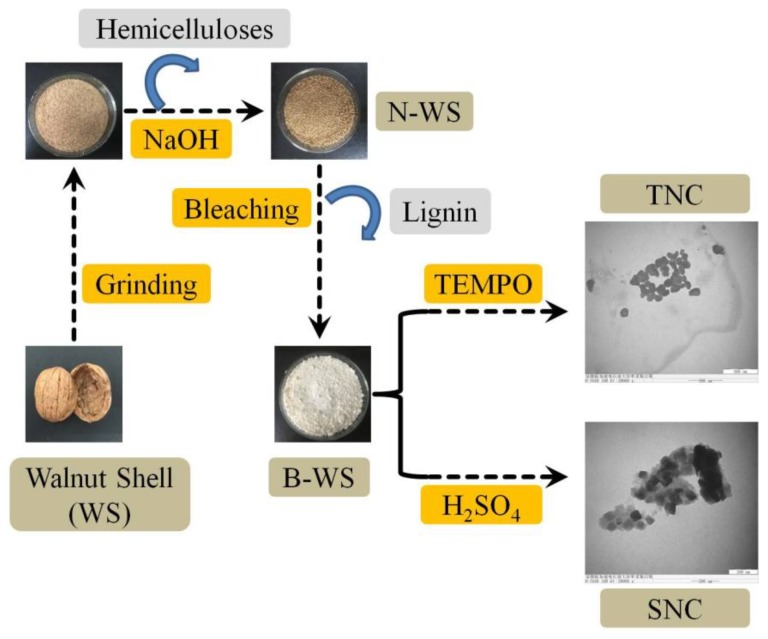
Procedure used to isolate nanocellulose from the walnut shell.

**Figure 2 polymers-11-01130-f002:**
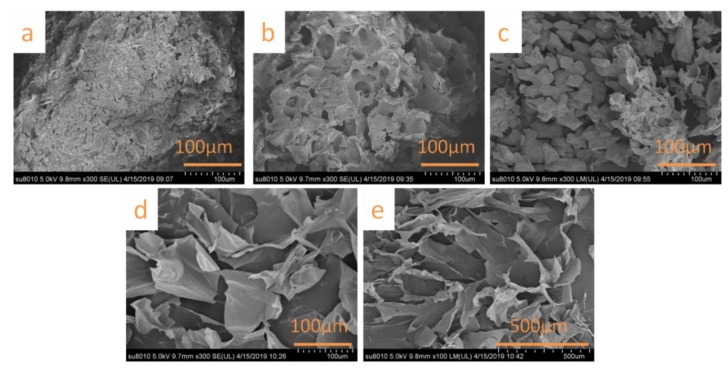
SEM micrographs of (**a**) WS, (**b**) N-WS, (**c**) B-WS, (**d**) nanocellulose obtained via TEMPO oxidation (TNC), and (**e**) nanocellulose obtained via sulfuric acid hydrolysis (SNC).

**Figure 3 polymers-11-01130-f003:**
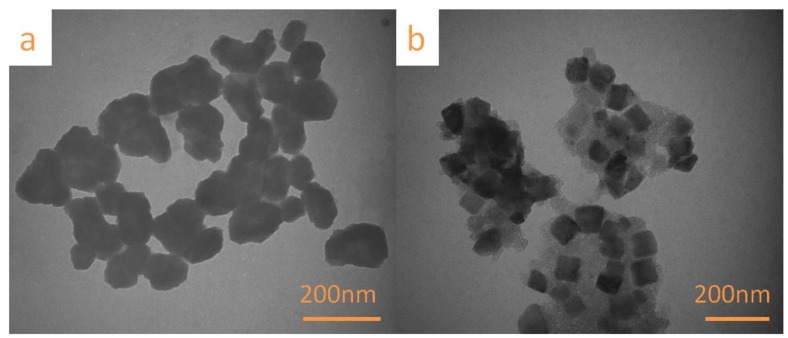
TEM images of (**a**) TNC and (**b**) SNC.

**Figure 4 polymers-11-01130-f004:**
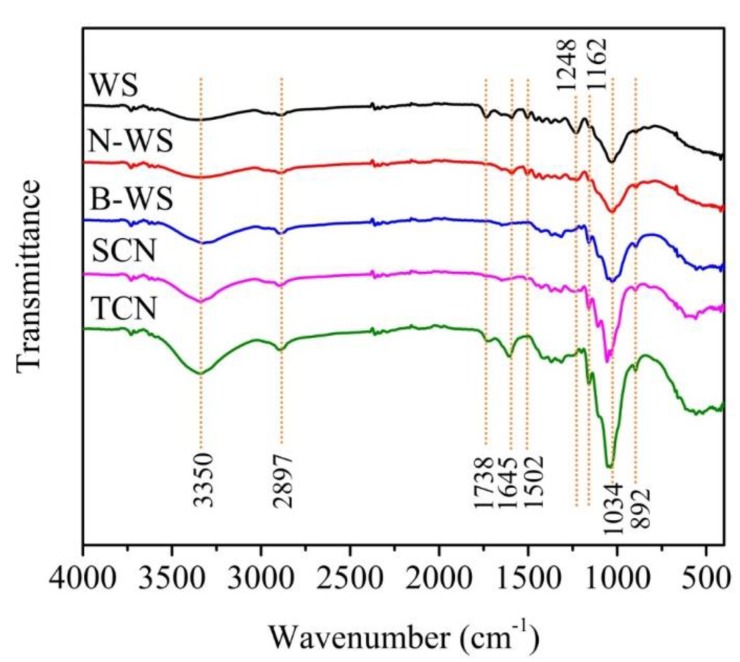
FTIR spectra of WS, N-WS, B-WS, TNC, and SNC.

**Figure 5 polymers-11-01130-f005:**
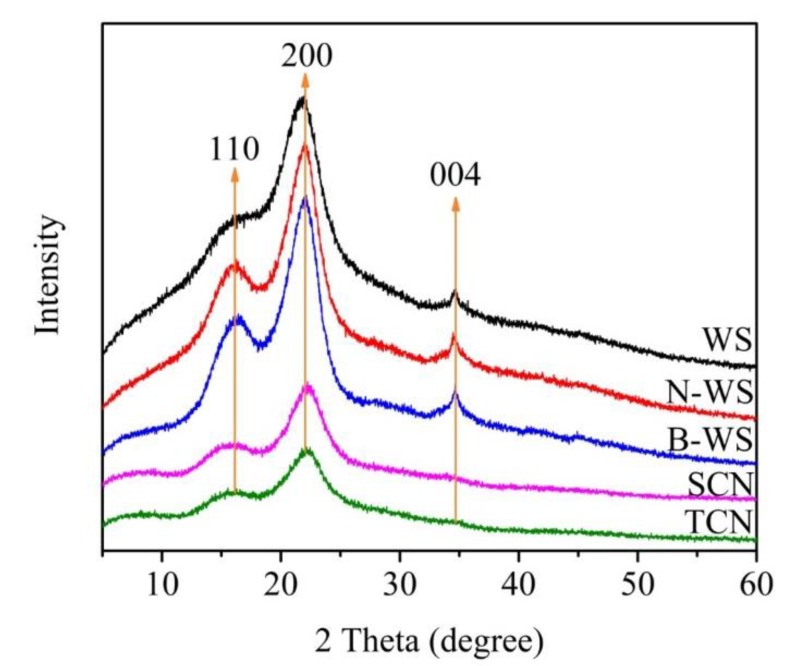
XRD patterns of WS, N-WS, B-WS, TNC, and SNC.

**Figure 6 polymers-11-01130-f006:**
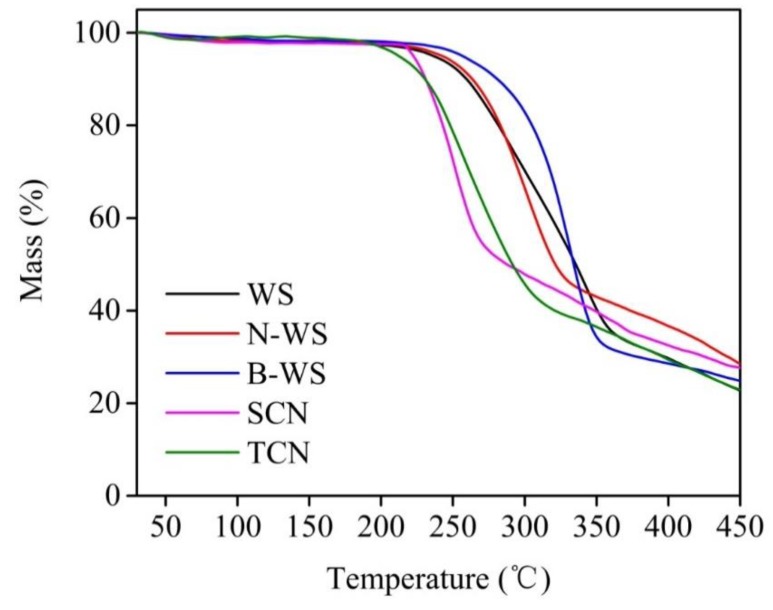
TGA curves of WS, N-WS, B-WS, TNC, and SNC.

**Figure 7 polymers-11-01130-f007:**
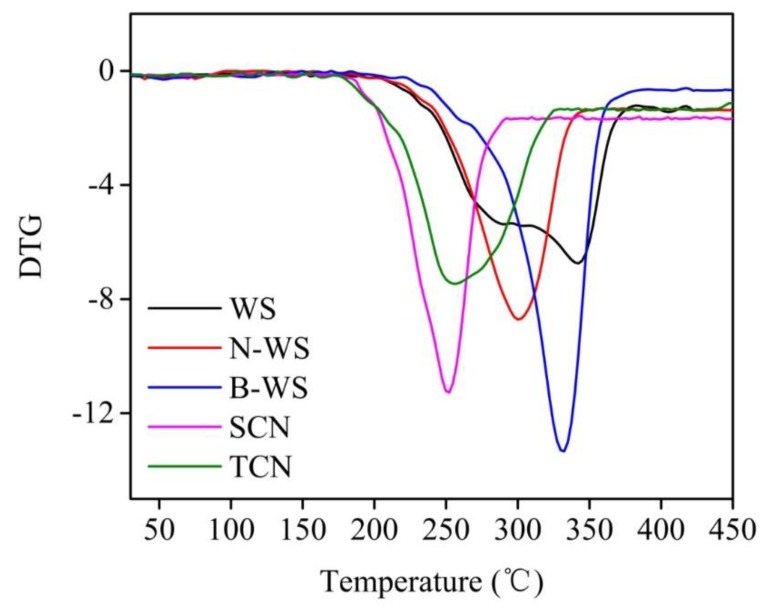
DTG curves of WS, N-WS, B-WS, TNC, and SNC.

**Table 1 polymers-11-01130-t001:** Chemical components of walnut shell WS, walnut shell treated by NaOH (N-WS), and walnut shell after bleaching (B-WS).

Sample	α-Cellulose (%)	Lignin (%)	Hemicelluloses (%)	Ash (%)	Benzene/Ethanol Extractives (%)
WS	27.4	36.31	31.3	3.6	1.57
N-WS	56.6	30.98	7.6	1.97	1.16
B-WS	87.9	0.17	1.8	1.64	0.41

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
