# Peer review of "Isolation and Characterization of Nanocellulose with a Novel Shape from Walnut (Juglans Regia L.) Shell Agricultural Waste"

_polymers, 2019, doi:10.3390/polym11071130_

Reviewer 1 Report

The paper under review “Isolation and characterization of nanocellulose with a novel shape from walnut (Juglans regia L.) shell agricultural waste” deals with preparation of nanocellulose from walnut by oxidation treatment.

General comments:

·        Authors should clearly indicate, how different “shape” (as stated in the title) influences the final properties and usage of such a nano-structures. Such a comparison with data given in literature should be emphasized.

·        It could be noted that whole procedures used in this Article were already applied and described, so the only novelty relay in using of walnut and preparation of nanocellulose with particular shape. Authors do not indicate what is the benefit of nanocellulose obtained by them over other nanocelluloses obtained by others?

·        Give some examples of possible (or already existing) usage of nanocellulose with different shapes

·        What “ensure the uniformity of the reaction” (line 119) mean?

·      Line 127: were there any experiments performed by Authors or by others regarding evaluation of time influence on NaOH treatment efficiency?

·        Lines 130-137: where all functional groups in oxidized B-WS converted into -COOH? Please evaluate a percent of conversion. Are there residual aldehyde groups visible on FTIR spectra?

Particular comments:

·        Introduction: please state shortly what is the benefit of using nanocellulose over cellulose, are there any differences in properties between them (besides size)?

·        Lines 50-63: add information about average content of cellulose in walnut

·        Line 79: “dispensability” – “dispersibility”?

·        Line 87: “yielding TCN) and sulfuric acid hydrolysis followed by ultrasonication (SCN).” – “yielding TCN) or sulfuric acid hydrolysis followed by ultrasonication (SCN).”

·        Line 98: “chlorite” – “chloride”

·        Lines 158-160: add reference

·        Lines 173-176: add number of scans, resolution, type of ATR crystal

·        Lines 181-182: different font

·        Describe shortly data presented in Table 1

·        Discuss differences observed in SEM images between TCN and SCN (see different magnification)

·        Lines 216-218: add reference

·        Lines 264-265: “There were 264 no significant differences between the TCN and B-WS XRD curves”: TCN curve resembles rather SCN one than B-WS

·        Line 274: “adsorbed or bound water from” – only slightly bound water

Author Response

Responds to the reviewer 1 comments

General comments:

1. Comment: Authors should clearly indicate, how different “shape” (as stated in the title) influences the final properties and usage of such a nano-structures. Such a comparison with data given in literature should be emphasized.

Response 1: Lines 229-231: The corresponding information has been added.

2. Comment: It could be noted that whole procedures used in this Article were already applied and described, so the only novelty relay in using of walnut and preparation of nanocellulose with particular shape. Authors do not indicate what is the benefit of nanocellulose obtained by them over other nanocelluloses obtained by others?

Response 2: Lines 231-232: It has been indicated in 3.3.

3. Comment: Give some examples of possible (or already existing) usage of nanocellulose with different shapes.

Response 3: Lines 231-232: Examples have been listed in 3.3.

4. Comment: What “ensure the uniformity of the reaction” (line 119) mean?

Response 4: Line 121: It has been revised to “ensure the reaction occurs evenly”.

5. Comment: Line 127: were there any experiments performed by Authors or by others regarding evaluation of time influence on NaOH treatment efficiency?

Response 5: The time influence has been studied by Saito, T. and Isogai, A.(2004). “TEMPO-mediated oxidation of native cellulose. The effect of oxidation conditions on chemical and crystal structures of the water-insoluble fractions”.

https://pubs.acs.org/doi/10.1021/bm0497769

6. Comment: Lines 130-137: where all functional groups in oxidized B-WS converted into -COOH? Please evaluate a percent of conversion. Are there residual aldehyde groups visible on FTIR spectra?

Response 6: Lines 136-137: The carboxylate content of carboxylated B-WS is 1.12 mmol/g as determined by conductivity titration.

Particular comments:

1. Comment: Introduction: please state shortly what is the benefit of using nanocellulose over cellulose, are there any differences in properties between them (besides size)?

Response 1: lines 39-41: It has been added in “Introduction”.

2. Comment: Lines 50-63: add information about average content of cellulose in walnut

Response 2: lines 63-64: The cellulose content of walnut has been added.

3. Comment: Line 79: “dispensability” – “dispersibility”?

Response 3: Line 81: The “dispensability” has been corrected to “dispersibility”.

4. Comment: Line 87: “yielding TCN) and sulfuric acid hydrolysis followed by ultrasonication (SCN).” – “yielding TCN) or sulfuric acid hydrolysis followed by ultrasonication (SCN).”

Response 4: Lines 89-90: The sentence has been modified according to the comment.

5. Comment: Line 98: “chlorite” – “chloride”

Response 5: Line 100: Here, the chemical used was NaClO2 (sodium chlorite) rather than NaCl (sodium chloride).

6. Comment: Lines 158-160: add reference

Response 6: Line 163: Reference has been added.

7. Comment: Lines 173-176: add number of scans, resolution, type of ATR crystal

Response 7: Lines 177-179: Number of scans, resolution and type of ATR crystal has been added.

8. Comment: Lines 181-182: different font

Response 8: Lines 184-185: The font size of lines 181-182 has been modified according to the text.

9. Comment: Describe shortly data presented in Table 1

Response 9: Lines 199-205: The discussion of chemical components has been added in 3.1, marked with red color.

10. Comment: Discuss differences observed in SEM images between TCN and SCN (see different magnification)

Response 10: There were no significant differences in SEM images between TNC and SNC. The SEM images of TNC and SNC showed similar structure which was caused by self-assemble of strong hydrogen bonding during freeze-drying.

11. Comment: Lines 216-218: add reference

Response 11: Lines 226-228: References have been added.

12. Comment: Lines 264-265: “There were 264 no significant differences between the TCN and B-WS XRD curves”: TCN curve resembles rather SCN one than B-WS

Response 12: Lines 278-280: The sentence has been revised and changed to “There were no significant differences in the position of X-ray diffraction peaks between the TNC and B-WS XRD curves”.

13. Comment: Line 274: “adsorbed or bound water from” – only slightly bound water

Response 13: Line 289: The sentence has been corrected according the comment.

Reviewer 2 Report

The aim of the research was to demonstrate the suitability of walnut shells as the raw material for the production of nanocellulose materials. In the introduction, Researchers aptly describe issues related with walnut shells - the agricultural waste produced in large amount every year. Thus, it was justification for carried out this research. Developed methods of preparation of nanocellulose was presented step-by-step. Conclusions are in accordance with provided results. Nevertheless, some points needed attention:

- please use the same abbreviation along whole manuscript. Currently, in abstract and conclusions there is “TNC”, while in remaining parts of the manuscript (including figures) there is “TCN”. The same applies to the “SNC” and “SCN”.

-page 3, lines 99 and 100 – the information about grade of chemicals was repeated.  

-line 181 – the font size is larger than in the rest of the text.

- The chemical components of WS were analyzed according to the Technical Association of Pulp and Paper Industry standards and the percentage composition was provided in Table 1. However, the brief comment about composition of WS should be added in part "3.1. Chemical components".

Author Response

Responds to the reviewer 2 comments

1. Comment: please use the same abbreviation along whole manuscript. Currently, in abstract and conclusions there is “TNC”, while in remaining parts of the manuscript (including figures) there is “TCN”. The same applies to the “SNC” and “SCN”.

Response 1: The abbreviations have been corrected to “SNC” and “TNC” the whole text, respectively.

2. Comment: page 3, lines 99 and 100 – the information about grade of chemicals was repeated.

Response 2: Lines 102-103: The repeated information has been revised.

3. Comment: line 181 – the font size is larger than in the rest of the text.

Response 3: Lines 184-185: The font size of lines 181-182 has been modified according to the text.

4. Comment: The chemical components of WS were analyzed according to the Technical Association of Pulp and Paper Industry standards and the percentage composition was provided in Table 1. However, the brief comment about composition of WS should be added in part "3.1. Chemical components".

Response 4: lines 199-205: The discussion of chemical components has been added in 3.1, marked with red color.

Round  2

Reviewer 1 Report

Being under review revised manuscript was substantially improved according to the reviewer’s comments thus it is worth to publish in a present form.

I have only found one mistake (line 231 “Nanocellulose with spherical or square sturcture" should be “structure”).

Author Response

Dear Reviewer,

Thanks a lot for your suggestion!

The response is listed:

1. Line 231: it has been coreected to "structure"

Sincerely,

Dingyuan Zheng

Reviewer 2 Report

In my opinion manuscript was revised accordingly and revised version could be published in a present form.

Author Response

Dear Rewiewer,

Thanks a lot for your suggestions! 

Sincerely,

Dingyuan Zheng